

# Technical comment on the paper of Dessert et al. "The dark matter interpretation of the 3.5 keV line is inconsistent with blank-sky observations"

Alexey Boyarsky[1], Denys Malyshev[2], Oleg Ruchayskiy[3] and Denys Savchenko[4,5*]

**1** Lorentz Institute, Leiden University,
Niels Bohrweg 2, Leiden, NL-2333 CA, The Netherlands
**2** Institut für Astronomie und Astrophysik Tübingen,
Universität Tübingen, Sand 1, D-72076 Tübingen, Germany
**3** Niels Bohr Institute, Copenhagen University,
Blegdamsvej 17, Copenhagen, DK-2100, Denmark
**4** Bogolyubov Institute for Theoretical Physics,
Metrolohichna Str. 14-b, 03143, Kyiv, Ukraine
**5** Kyiv Academic University, 36 Vernadsky blvd., Kyiv, 03142, Ukraine

★ dsavchenko@bitp.kiev.ua

## Abstract

An unidentified line at energy around 3.5 keV was detected in the spectra of dark matter-dominated objects. Recent work [1] used 30 Msec of XMM-Newton blank-sky observations to constrain the admissible line flux, challenging its dark matter decay origin. We demonstrate that these bounds are overestimated by more than an order of magnitude due to improper background modeling. Therefore, the dark matter interpretation of the 3.5 keV signal remains viable.

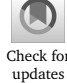
An X-ray line at $E \simeq 3.5$ keV has been found in 2014 [2, 3]. Many consistency checks, as well as follow-up detections, have been reported, while non-detections have not ruled out the dark matter interpretation of the signal (see [4] for review). Ref. [1] (**DRS20** in what follows) recently reported bounds on the decay lifetime, which are about an order of magnitude below those required for dark matter interpretation. The improvements of these bounds compared with the previous results are much stronger than one would expect based solely on the increase of the exposition. We demonstrate that such a strong increase is mainly an artifact of overly restrictive background modeling.

We use 17 Msec of *XMM-Newton* MOS observations pointing $20° − 35°$ off the Galactic Center.[1] This dataset contains 57% of the total exposure of **DRS20**, including 503 of their 534 observations. Thus, we expect the flux upper limit to differ by $\approx \sqrt{30/17} \approx 1.32$. Different dark matter profiles are consistent with each other in this region making the limits more robust.

---

[1]For the list of observations see [5].

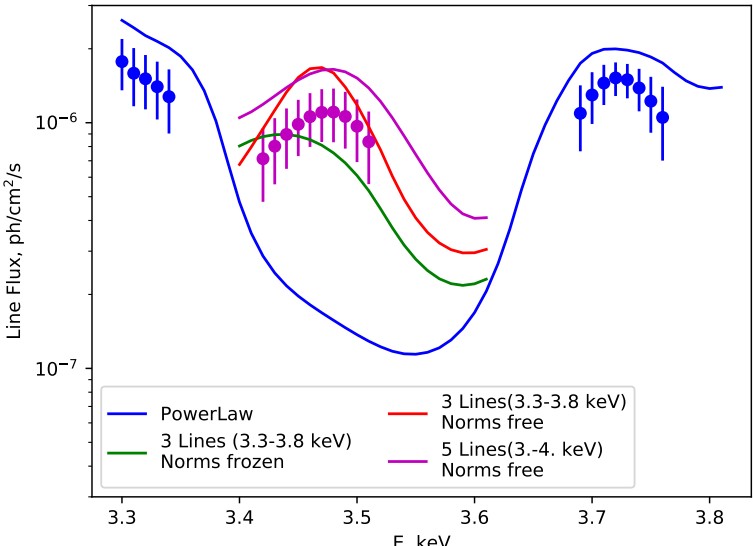

Figure 1: Upper limits (95% CL) on the extra line flux against the models, described in Table 1 (colors coincide with the model names). Data points show the energies where the lines are detected at more than $3\sigma$ level over the continuum of the corresponding color (errorbars are $\pm 1\sigma$).

Table 1: Four background models for line search.

| Model name | Model components | Interval | $\chi^2$/dof | Line at $E_{\text{fid}}$? | 95%CL at $E_{\text{fid}}$ $10^{-6}$ cm$^{-2}$s$^{-1}$ |
|---|---|---|---|---|---|
| **Blue** | Powerlaw | 3.3-3.8 keV | 88.76/96 | No | 0.16 |
| **Green** | Powerlaw Line @ 3.3 keV Line @ 3.68 keV | 3.3-3.8 keV | 68.86/96 | No (0.8$\sigma$) | 0.70 |
| **Red** | Powerlaw Line @ 3.3 keV Line @ 3.68 keV | 3.3-3.8 keV | 68.86/94 | No (1.3$\sigma$) | 1.41 |
| **Magenta** | Powerlaw Line @ 3.12 keV Line @ 3.3 keV Line @ 3.68 keV Line @ 3.9 keV | 3.0-4.0 keV | 163.0/193 | Yes (4.0$\sigma$) | 1.72 |

*Our results are shown in Fig. 1, while Table 1 summarizes the details of our modeling and shows 95% CL at fiducial energy $E_{\text{fid}} = 3.48$ keV.*

First, we searched for a narrow line atop of a folded powerlaw (plus an instrumental continuum fixed at high energies) across the interval 3.3-3.8 keV. Our limits ("blue model") are consistent with **DSR20**: strong constraints around 3.5 keV, lines at $\sim$ 3.3 and 3.68 keV detected with significance $\geq 3\sigma$, consistently with significant weakening on the **DRS20** limits at these energies. Such lines (Ar XVIII and S XVI complexes around 3.3 keV, and Ar XVII plus K XIX around 3.68 keV) are detected in astrophysical plasma both in galaxy clusters [2,6,7] and in our Galaxy [8–11]. Besides, the presence of the weak instrumental lines – K K$\alpha$ at 3.3 keV

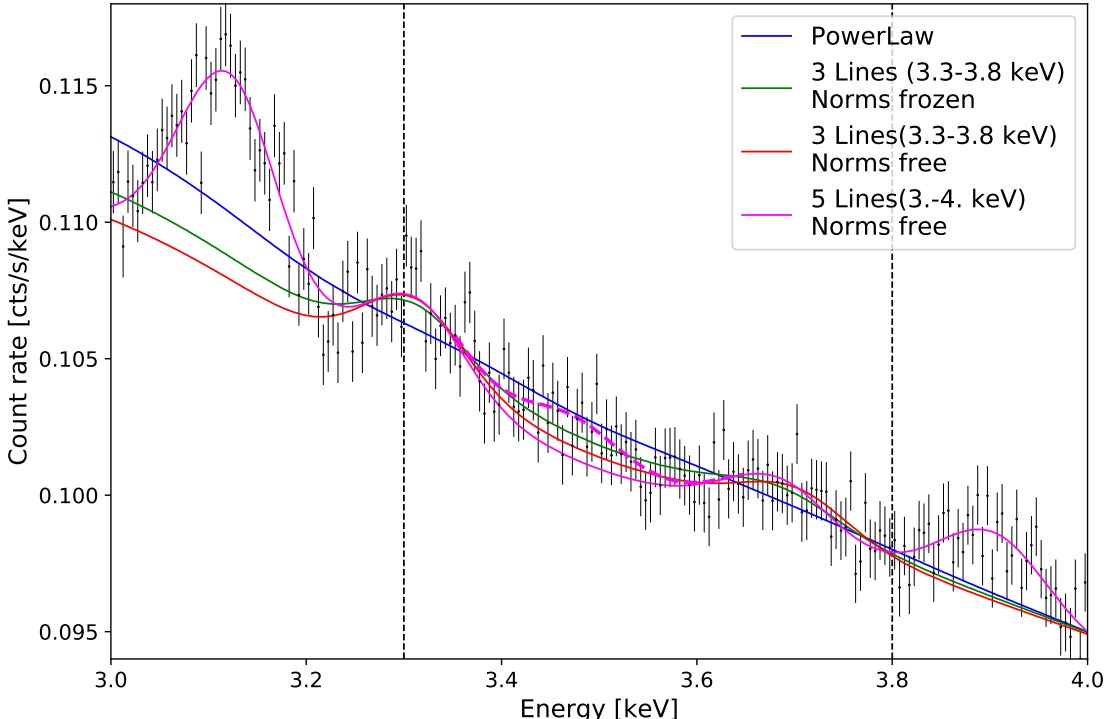

Figure 2: Shown are the actual spectra and background models, described in this Comment. The necessity for the additional astrophysical lines above the continuum powerlaw is clearly seen. One can see that the level of the background continuum decreases with the addition of astrophysical lines. The dashed magenta line shows the model with the line at $E_{\text{fid}}$ included. Vertical dashed lines correspond to the modeling range of "blue", "green", and "red" models.

and Ca K$\alpha$ at 3.7 keV has been reported, see [11] and refs. therein.

Therefore, next we add to the model extra Gaussians at 3.3 keV and 3.68 keV ("green model", Table 1) and repeat the analysis in the 3.3-3.8 energy range. The bounds weaken by a factor $\sim 4$ (Fig. 1, green line) at $E_{\text{fid}}$. This weakening is consistent with Fig. S14(A) of **DRS20**. A background model without these lines raises the powerlaw continuum, which artificially lowers the upper limit on any line (see [12] for the previous discussion). Indeed, Fig. S16(B) of **DRS20** demonstrates that the bestfit value of the line flux at 3.5 keV, parametrized by $\sin^2(2\theta)$, is negative at the level $\sim -1.5\sigma$ – background is over-subtracted. We notice that the normalization of two lines at the end of the interval was fixed to their best-fit value during this procedure.

When, instead, we let the normalization of the lines vary freely, while adding an extra line around 3.5 keV (as in [2, 3, 9, 12]) – the upper limit on the flux weakened by an extra factor of $\sim 2$ ("red model").

Interval 3–4 keV contains two more known lines – Ca XIX complex around 3.9 keV and Ar XVII plus S XVI around 3.1 keV *c.f.* [2, 9, 11]. We repeat our analysis in this interval with two extra lines in the model ("magenta model"). We find a $4\sigma$ line at $E_{\text{fid}}$ and the upper limit weakens accordingly. Further extending the model to 2.8–6.0 keV, carefully modeling all astrophysical and instrumental lines and accounting for all significant line-like residuals, does not change the results.

The described above models are shown in 3 – 4 keV energy band in Fig. 2 with corresponding colours. Fig. 2 illustrates that the model with no astrophysical lines fails to ade-

quately model the data on a broader energy range, and overestimates the continuum level, as discussed above.

## Conclusion

We demonstrate that the constraints from long exposure blank-sky observations [1] strongly depend on the background model. Namely, proper inclusion of the line complexes at 3.3 keV and 3.68 keV relaxes the bound by a factor $\sim 8$. The extension of the fitting interval to 3–4 keV weakens the bound by more than an order of magnitude (magenta line in Fig. 1 compared to the blue line, reproducing **DRS20**) and also leads to the detection of the line at 3.5 keV at $4\sigma$.

**DRS20** investigates the effects of these lines in the *Supplementary Material*. However they *(i)* fix normalization of their best fit background values, reducing their effect (reproduced by our "green" model) and *(ii)* chose to report more stringent bounds as their final result.

When claiming *exclusions*, one should be careful to push all systematic uncertainties in the conservative directions. In this particular case, to claim the strongest "powerlaw" bound (as done in **DRS20**) one should *prove* that other known lines are not present in a particular dataset. Moreover, if the analysis at a wider energy interval (3–4 keV) gives weaker constraint, we see no reason not to report it as a proper conservative bound.

Furthermore, to interpret the exclusion in terms of the decaying dark matter lifetime, one needs to adopt the most conservative density profile. In particular, the local dark matter density was adopted in **DRS20** to be $0.4\,\mathrm{GeV/cm^3}$. It has a systematic uncertainty of a factor $2-3$ [13], see also discussion in [14], which should be propagated into the final conservative bound.

The spectral resolution of modern X-ray satellites is below that, required to resolve the intrinsic shape of astrophysical or putative dark matter decay lines. Future X-ray spectrometers will be able to finally settle this question.

## Other comments

Below we comment on other inconsistencies in **DRS20**. The above conclusions are not based on them.

**PN out-of-time events not subtracted?** For the PN camera the out-of-time events were not subtracted. Indeed, the scripts `dl2dat.sh` and `spc2dat.py` of **DRS20** show that count rates from files `pn*-obj.pi`, produced by the ESAS script `pn-spectra` were used. Instead, out-of-time subtracted spectra, produced by `pn_back` (filename pattern `pn*-obj-os.pi`) should have been used, according to the ESAS manual.

**Wrong PN count rate?** Fig. 2 of **DRS20** shows that counts rates of stacked MOS and PN spectra are similar. However, it is known that count rate of the PN camera is $\approx 3$ times higher than of the MOS cameras, *c.f.* [15, Fig. 7 & Table 2] or [12, Fig. 1]. This difference is not explained in the text. **DSR20** showed the count rate for the PN camera of `ObsID 0653550301` (Fig. S11) which we reproduced. The MOS count rate for the same observation is a factor of $\sim 3$ lower.

# Acknowledgements

The authors acknowledge communication with K. N. Abazajian and D. Iakubovskyi. This project received funding from the European Research Council (ERC) under the European Union's Horizon 2020 research and innovation programme (GA 694896) (AB and OR). The work of DM was supported by DFG through the grant MA 7807/2-1. OR acknowledges support from the Carlsberg Foundation. The authors acknowledge support by the state of Baden-Württemberg through bwHPC.

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
