# Peer review of "Technical comment on the paper of Dessert et al. "The dark matter interpretation of the 3.5 keV line is inconsistent with blank-sky observations""

_SciPost Astronomy Core, doi:SciPost Astro. Core 1, 001 (2021)_

## Round 1 · Referee Report · Dominique Eckert (Referee 1) · 2020-10-27

Strengths

The issue of whether an unidentified emission line at 3.5 keV exists is a very important one, as this possible line has been reported (at low significance) in numerous previous papers and, if confirmed, it could become the "smoking gun" for sterile neutrinos as a candidate for dark matter. The paper that is criticized here claims to close down the issue by presenting upper bounds on the sterile neutrino mixing angle that are much lower than the claimed detections. If this particular paper was flawed it is important for the community to be aware of it.

Weaknesses

The paper claims there are additional lines, both instrumental and astrophysical, around the energy range of interest. Modelling these additional emission lines weakens the upper bounds on the sterile neutrino mixing angle by about an order of magnitude. However, I find that the evidence for the existence of these lines is not presented in a compelling way. The only figure of the paper (Fig. 1) is confusing and hard to read, since the full spectrum is not presented. The authors need to make a better job at convincing the reader that the addition of these lines is required by the data; formally speaking, all the fits presented in Table 1 are acceptable.

Report

The question of whether such a paper is suitable for publication in a refereed journal is a debated one. I personally think that disagreement in the scientific community over some contentious issues is normal, and cross-validation should be encouraged, so I am personally inclined to be positive about this type of papers.

Requested changes

I would really like to see the authors showing a full spectrum including a broader energy range. At face value, Fig. 2 of the DRS paper looks extremely compelling, but I fully understand that if some additional lines are present, the spectral resolution of the XMM CCDs is not sufficient to distinguish them from a power law in such a narrow energy range. However, if that is the case, the residuals associated with the wrong model should be clearly visible when looking at a broader spectral range. To make their point more clearly, I would like to advise the authors to add a new figure in which the full stacked spectra are presented and the difference between the DRS model and the more complex model advocated here is shown.

---

## Round 2 · Referee Report · Dominique Eckert (Referee 1) · 2021-4-12

Report

Dear colleagues,
Thanks for addressing my previous points, I think the new figure showing the actual spectrum is very nice and it convincingly makes the case for including the various lines. I am happy to recommend acceptance of the paper.
Cheers,
Dominique

---

## Round 2 · Author Response

Dear Editor

We are submitting the revised version of the Technical comment with changes suggested by the Referee.
These changes aimed at visual clarification for the readers of the points in the article.
It took more time than expected for a minor revision due to COVID-related disruptions.

Sincerely,
The Authors

---

## Round 2 · List of Changes

1. Fig. 2 added, which shows the actual spectra together with models under consideration.
  2. Additional paragraph referencing Fig. 2 added.
  3. Typesetting changed to match SciPost style.

---

## Editorial Decision

published